# A Case Study of a Digital Data Platform for the Agricultural Sector: A Valuable Decision Support System for Small Farmers

**Juan D. Borrero \*** and **Jesús Mariscal**

Agricultural Economics Research Group, Department of Management and Marketing, University of Huelva, 21002 Huelva, Spain; jesus.mariscal@dem.uhu.es
\* Correspondence: jdiego@uhu.es

**Abstract:** New players are entering the new and important digital data market for agriculture, increasing power asymmetries and reinforcing their competitive advantages. Although the farmer remains at the heart of agricultural data collection, to date, only a few farmers participate in data platforms. Despite this, more and more decision support systems (DSSs) tools are used in agriculture, and digital platforms as data aggregators could be useful technologies for helping farmers make better decisions. However, as these systems develop, the efficiency of these platforms becomes more challenging (sharing, ownership, governance, and transparency). In this paper, we conduct a case study for an accessible and scalable digital data platform that is focused on adding value to smallholders. The case study research is based on meta-governance theory and multidimensional multilayered digital platform architecture, to determine platform governance and a data development model for the Andalusian (Spain) fruit and vegetable sector. With the information obtained from the agents of this sector, a digital platform called farmdata was designed, which connects to several regional and national, and public and private databases, aggregating data and providing tools for decision making. Results from the interviews reflect the farmer's interests in participating in a centralized cloud data platform, preferably one that is managed by a university, but also with attention being paid toward security and transparency, as well as providing added value. As for future directions, we propose further research on how the benefits should be distributed among end users, as well as for the study of a distributed model through blockchain.

**Keywords:** digital platform; data governance; data ownership; Big Data; agriculture; decision support system

## 1. Introduction

The agricultural sector is undergoing a period of rapid transformation, driven by global impacts from climate change, demographic and migratory flows, and uneven economic growth [1]. In this context, digitization can help the agricultural sector to increase crop yields while contributing to the fight against climate change. Although the farmer remains at the heart of agricultural data collection, he is sometimes unable to access and manage it easily. In addition, new actors that are raising power asymmetries are entering into this digital market, such as the suppliers of inputs and machinery for agriculture, and large software or industrial companies [2].

Digital technologies, such as the Internet of Things (IoT) and machine learning (ML) techniques offer new opportunities for agriculture [3,4]. These technologies involve data sharing, homogenization, editability, reprogrammability, and distribution [5–7], meaning that no single actor owns the ownership of the digital platform [8]. A key element in creating value with the use of these technologies is data [9–12]. The question of how value is created for agriculture from data is only beginning to be addressed [13]. Studying this issue requires the consideration of both the technical and economic properties of the data [14]. Private data that is provided by farmers describe yield, farm weather conditions,

or plant condition. In parallel, information that is collected or produced by the public sector and made freely available for reuse for any purpose, such as satellite images (e.g., Copernicus) and reports (e.g., National Agricultural Statistics Services), will also enable the economic growth of the agricultural sector. Data is also available from social media platforms (e.g., Twitter) that allows for the analysis of user behavior. The power of data could empower farmers in their relationships with suppliers and customers, but they would only be willing to voluntarily share data as long as their individual benefits exceed the cost of sharing [15]. Additional factors such as security, transparency, intentions, stakeholder decisions, and organizational rules place limits on the value created from data [13,16].

Much research analyzes the use of digital agricultural technologies with new decision-making tools; for example, in input purchasing, farm management, and product pricing [17–20]. However, we also believe that insufficient attention has been paid towards the implications of Big Data for small agriculture [21,22], and to date, only a few farmers participate in Big Data platforms [23,24].

Digital platforms as ecosystem of actors who share a common data-related goal, and of systems that uses data to produce value and incentivize actors to engage in valuable interactions [7], can change the economic sector where they operate [25]. Digital platforms combine and deploy analytics and cloud computing solutions within a competitive market [25,26]. They may also require public investments such as general digital infrastructure, and services such as cloud storage or connectivity [27].

Although agricultural digital platforms are not explicitly defined in the literature, they are generally treated as co-creation, in the sense that they focus on a process of adding value to data that includes both an agricultural decision support system (DSS) and a cloud platform with ML algorithms [28]. While an agricultural DSS is very useful in helping farmers to perform various agriculturally smart decisions [29–31] such as smart irrigation decisions [32], crop nutrient management [33], or feeding decision making [34], a cloud platform with ML algorithms is the brain of the platform, taking care of data organization and processing [35]. According to this scheme, actors (users) co-create value without distinctions between the roles and the actions between the service provider and the data or service user [36].

There are still numerous issues and challenges in the use of digital technologies in agriculture [37,38]. First of all, data ownership [22,38]: Although farmers provide data, they have no control over the use of the data [2]. Second, privacy concerns cause farmers to be reluctant to share data [39,40]. Transparency in data use [12,16] and the way in which benefits are distributed [12] are also unclear. Finally, the question of how to govern digital platforms has also been a topic of ongoing study [41].

The case study aims to address the above-mentioned research gap, the issues, and the challenges of digital platforms as useful DSSs for smallholder farmers. Within the framework of the "AgroMIS singular project: ceiA3 strategic instrument towards a Modern, Innovative and Sustainable Agri-food productive fabric: engine of the Andalusian rural territory [42]" (hereinafter, AGROMIS I) financed with FEDER funds, where several Spanish projects have been launched in accordance with the Research and Innovation Strategy for the Smart Specialization Platform [43] (RIS3), two research questions (RQs) are proposed:

- RQ1. What are the necessary elements that an Agriculture Data Platform (hereinafter, AgDP) should have to support horticultural agriculture? The purpose of this question is to identify and to select technical elements that should be included in a fruit and vegetable data platform, due to the current lack of systematic empirical analysis of digital platform research in agriculture.
- RQ2. What should be the governance model for this AgDP? The objective is to identify the key elements that guarantee data ownership and security, as well as the sustainability and growth of the platform.

To answer these questions, we first conduct a research study based on meta-governance theory [44], which, although it has not been used in the context of digital platforms, will

allow us to analyze the interests and relationships of the various stakeholders, and then develop a case study for a data aggregator platform.

The rest of this paper is organized as follows: The literature section provides an overview of the digital technologies and platforms for agriculture that will serve as the basis for the study design described in the methodology section, and that will be applied to the case study of a digital platform and its governance model for the Andalusian fruit and vegetable sector. Finally, the results are discussed, and conclusions and future challenges are provided.

## 2. Literature Review

We refer to digital technologies as an infrastructure with Internet access, electricity supply, and the use of smartphones and their application for agricultural purposes by users with a certain level of accessibility, training, and support [45]. Using the distinction between embedded and non-embedded innovations [46], digital technologies can be separated into those that are embedded in physical devices such as farm machinery or sensors; and non-embedded software tools such as farm advisory applications and online platforms. The first category belongs to so-called precision agriculture, which relies on a set of technologies that work together to enable data generation, collection, and analysis, such as remote sensing [47], wireless sensors [48], drones [49], or robots [29]. In the second category are software tools that run on mobile devices such as smartphones or tablets [50], and that can help farmers to manage their farms using private and public data. From the interrelation of both categories, other new services emerge in the field of collaborative economy [51], or cooperative work [52].

The above goods and services could be classified as public goods, private goods, common-pool resources, and club goods [53]. According to this, digital technologies are considered to be club goods if they are protected by patents, requiring users to pay a license fee for their hardware, and allowing the provider to use the agricultural data collected. In contrast, digital technologies that give non-farm-specific advice have the nature of public goods. As far as small-scale farmers are concerned, this farm advice can be considered as merit good, free or provided cheaply, because farmers are not willing to pay and the government wishes to encourage their consumption due the positive externalities associated with the data [54]. The data and the value created on a digital platform for agriculture can be treated as a private good [12], a club good [55], or a public good [56], depending on the use, the remuneration, and its users.

The market entry of new companies, and the shift from input-based business models to service-based business models of traditional companies reinforces their competitive advantages. Among them, we find [54]:

- Large multinational agricultural input companies, such as Corteva, Bayer, or Chem-china; or agricultural machinery such as John Deere, CNH, or AGCO, which can commercialize digital agricultural goods as complementary assets;
- Large multinational software companies outside the agricultural industry, such as IBM, Microsoft, TCS, Tencent, or Alibaba, which are interested in digital agricultural technology;
- Large industrial companies such as Bosch or XAG, which provide sensors or drones for precision agriculture;
- Start-ups, some of them coming from universities funded by venture capital investors or multinational input and technology companies.

We have found several digital farm data platforms currently operating under a variety of governance models (Table 1). Most of these systems provide farmers with information on farm agronomy and management based on the data uploaded by them. Some of these platforms are independent software companies such as FBN [57] or Farmobile [58], with varying degrees of openness to data and relationships with farmers, while others are owned by agricultural input companies such as Granular (Corteva) [59] or Climate Corporation (Bayer) [60]. Input companies may benefit from the complementarity between digital data services and the products or services that they sell to farmers. Similarly, farm

equipment companies, such as John Deere or CNH are also leveraging their relationships with customers to develop data platforms. Microsoft offers Azure "FarmBeats" [61], which is a cloud-based platform on the geographic information system (GIS) and the IoT system that enables data collection for agriculture. Finally, some Big Data platforms are promoted by administrations or managed by non-profit organizations, such as FaST [62] and Ag Data Coalition (ADC) [63].

**Table 1.** Main agriculture data platforms.

| Data Platform | URL | Region | Source | Governance Model |
| --- | --- | --- | --- | --- |
| FaST | https://fastplatform.eu/ (Accessed on 25 March 2022) | EU | Public | DG Agriculture and Rural Development |
| Climate Corporation | https://climate.com (Accessed on 25 March 2022) | EU | Private | Bayer |
| New Vision Coop | https://www.newvision.coop/ (Accessed on 25 March 2022) | USA | Private | Cooperative |
| Winfield Solutions | https://www.winfieldunited.com/ (Accessed on 25 March 2022) | USA | Private | Cooperative |
| ADC | http://agdatacoalition.org/ (Accessed on 25 March 2022) | USA | Private | Nonprofit organization |
| GISC | https://www.gisc.coop/ (Accessed on 25 March 2022) | USA | Private | Cooperative |
| JoinData | https://join-data.nl/ (Accessed on 25 March 2022) | EU | Private | Nonprofit organization |
| Farmers Business Network | https://www.fbn.com/ (Accessed on 25 March 2022) | USA | Private | Farmer's Business Network, Inc. |
| Azure Farmbeats | | USA | Private | Microsoft |

## 3. Materials and Methods

### 3.1. Problem Definition and Case Study Research

The aim of the research is to design and develop an accessible digital data aggregator as a starting point for a future useful digital platform; an intelligent system that is based on data integration and ML techniques, in order to provide decision support in agriculture in general, and in the horticulture sector in particular.

For the design and development of a platform to help farmers solve complex issues related to crop production and governance, the case study was guided using the approach of [44,64–66] in an iterative conceptual and empirical approach. The data development process for the agricultural sector can be broken down into four phases—data generation, collection, storage, and analysis [22]; here, a multidimensional multi-layered digital platform framework is proposed (Figure 1). This approach was particularly beneficial, due to the current lack of systematic empirical analysis for digital platform research in agriculture.

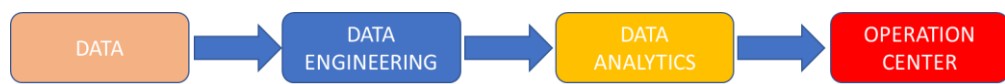

**Figure 1.** Basic digital platform system architecture.

Data is the main asset of the platform, a fungible good that is offered by the farmer in exchange for higher-value information. The platform must collect and store data to support value creation, where the total value generated will be the product of the benefit accrued by the data, the benefit itself generated by the system, and the benefit received by the user community [7]. Stakeholders will use the data that is developed by the system to achieve greater results for the same cost [67].

Qualitative data were collected using the framework in [65], through semi-structured interviews with experts (three researchers from the University of Huelva, Spain, a technician from the association of berry producers of Huelva, Freshuelva, and a technician from the IT

department of the Directorate General Agriculture of the Andalusian government) in two steps. In the first one, we worked on data value by asking the questions "For what purpose would you share the data with this platform?" and "To whom does the value generated from the data belong?".

The literature review and analysis of existing digital farming platforms served to identify key issues and to develop an initial list of questions that was validated by a pilot group of 10 berry growers from the province of Huelva, Spain. As a result, the wording was modified, with some words eliminated and others added. The final list served as a guide for the second phase of interviews, where an initial data platform design had already been shown.

### 3.2. Data Platform Design and Governance

The architecture of a digital platform has three layers [68], and it relies on three technologies [29]. In relation to the levels, these include the data that feed the platform; an infrastructure for data acquisition and storage, and computational algorithms; and a network composed of the participants and their relationships. The technologies that make a digital platform work are Big Data [19,21,22,24,46,69], ML [33,35], and cloud computing [19,48,70–72]. Big Data technology allows for the processing of data aggregated to the platform; recommendations and predictions can be performed using ML techniques, and cloud computing provides technical support for reliability and security in aggregated data.

In their simplest form, digital platforms use data to create value for users [7]. In some cases, users offer data for free in exchange for a share of the value created [73], while in others, the platform bears the costs and internalizes the benefits to achieve network externalities [74] that occur when these platforms bring together multiple and numerous user groups [75]. Network externalities, both as direct network externalities when more farmers join the platform, and indirect ones if more researchers or startups create services for the platform, are other key factors that adds value to the platform. A community exists because the data platform provides all users with a more effective mechanism for deriving benefit from the data than if they were to do so in isolation. The user community consists of all interested parties pursuing some data-related goal or action. This community, with strong and weak structural relationships between actors, includes both users that contribute data (farmers) or applications (complementors), as well as participants who simply provide data (government) or who access more complete information from aggregated data (citizen). Outlying stakeholders may include other companies in the agri-food sector, such as suppliers of goods and services, and public agencies [22]. In a more mature model, the beneficiary consumers (e.g., farmers) remunerate the services through payments, or are charged for their data.

The technical perspective also considers digital platforms as being software that are complemented by external services that can amplify the functionality of the platform [76]. Thus, another type of users, the complementors [77], use part of the resources, such as data, via APIs [78]. To ensure a valid value proposition, the platform needs both the consumer and the complementor, but neither party is willing to join, as long as the other is not populated [79].

The sociotechnical perspective, analyzing the platform as an ecosystem, focuses on how the platform integrates and governs a community of stakeholders [75]. Depending on the ownership archetype, whether it is a centralized platform, a consortium, or a decentralized network, they need to balance their control rights [75,78]. This approach focuses on how the stakeholders interact with each other and create value [80]. Platform ownership, in the sense of power distribution in the ecosystem that can be centralized or decentralized, is an essential attribute for the design and governance of a data platform ecosystem [76]. Thus, community organization, value allocation among stakeholders, and design requirements are determined by the business model adopted by the platform [81,82]. A poorly specified design, such as data development toward objectives that are not valued by actors, diminish the value that is created by the platform, and reduces network performance.

In meta-governance theory [44], the common use of the prefix "meta-" refers to the idea of "beyond", which could already provide a simple definition of the term meta-governance as being the "governance of governance". In the same vein, the idea of meta-governance comes from discussions regarding the reciprocal implication of relationships between hierarchical, market, and network models [44]. Thus, meta-governance presupposes the interaction of hierarchy in the management of interests, the coordination of networks and their multiple actors, and of the unpredictability of market forces. Therefore, meta-governance is not intended to be characterized as a model of super-governance, but as an approach that seeks to contribute to the resolution of the typical failures of each of the elements, models, and orders of governance and their combinations. It also seeks to establish a coherent, updated, and flexible set of values, principles, and norms, with the application of ethical standards and rules of conduct. This is so that the actors that are involved can adequately face their dilemmas and challenges in their respective spheres of power. From this perspective, the governance model for the platform contemplates the three attributes pointed out by [4] (a multiplicity of actors, complex structural relationships, and a governing governor) as being necessary for envisaging an organic and harmonious governance model, since there are often conflicts of interest or unclear technology development dynamics [44,83].

## 4. Results and Discussion

### 4.1. Agriculture Data Platform Elements

After an analysis of the data available at the time of platform development, the following data sets from public and private sources were used (Table 2).

**Table 2.** Agricultural datasets aggregated to the AgDP.

| Type of Source | Type of Data | Provider | Type of Aggregation |
|---|---|---|---|
| Satellite imagery | Public | National government | Link |
| Meteorological information | Public | National government | API |
| Crops | Public | Regional government | Csv |
| Prices and sales data | Public | Regional government | Json |
| Plant disease incidence data | Public | Regional government | Xml |
| Citizen data from social media platforms | Public or Private | Twitter | API |
| Apps | Private | University spin-off | API |

Public data were collected by public agencies, such as satellite imagery or weather conditions through sensors. Other public data was generated through administrative processes by them and provided in an anonymized form, as it contained private information. Data that might contain individual-level or other private information were grouped by zip codes or products.

Other key private data can be collected by individual farmers. For example, farm and crop level data are collected by UAVs, field sensors, cameras, or apps. This project has enabled collaboration with university spin-offs to include in the platform some applications for free and open use by farmers (see the farmdata.es and apps section), in exchange for sharing this data with the platform so that these spin-offs can develop valuable artificial intelligence (AI) services for farmers.

In the information technology (IT) field, ACID is often used to indicate atomicity, consistency, isolation, and durability. These are properties that relational databases such as MySQL, Oracle, SQL Server, and PostgreSQL bring to systems and make them more robust and less vulnerable to failures. NoSQL (Not Only SQL) database management systems such as MongoDB, Cassandra, or Redis are very useful when you do not have an exact schema of what you are going to store, and for massively data processing [35]. Likewise, when you have a massive amount of data and want to access it, you can also use Hadoop,

which is a software framework or a set of programs, and open source code. For the case study between the consistency of SQL or the high availability of NoSQL or Hadoop, the former was preferred because the AGROMIS I project starts with scarce economic resources and data, it does not need availability and real-time processing of data, and the possibility of scaling is always feasible.

The idea of data aggregation is simple: all of the farm data collected by farmers, as well as the public data on climatic conditions, plant diseases, and market conditions, can be aggregated into a single space and made available in an aggregated form for information queries. There are two key elements that the data aggregation service should comply with: the updating of this information should be as automatic and accurate as possible, since adjusting to reliable and current data is key when we are talking about predictive systems; and to gather a large number of farms. To comply with these elements, as well as to ensure security and reliability, the aggregator system uses Amazon Web Services (AWS) [84], which is the most widely adopted and complete cloud platform in the world [85], and it encompasses a large number of services to perform different types of activities, from storage to computing.

Apache Spark (along with Hadoop) seems to be the perfect candidate framework if we want to use it to make predictive models with large amounts of data. However, as the AGROMIS I was not a Big Data project at the beginning, we opted for an analytical Data Lake [86], that is, a centralized storage repository containing data from various sources in their original and unprocessed format. This makes it possible to conserve data, creating repositories with large volumes of information without investing a lot of time in structure, and with the possibility of cost-effective expansion. Instead of opting for an on-premises data lake with haddoop, the platform chose a data lake in the cloud with AWS, as it has the following advantages:

- Easier and faster to initialize, allowing it to start gradually;
- More economical, with a pay-per-use model;
- Easier to scale as needs increase, eliminating the stress of calculating requirements and obtaining approvals.

The system architecture and its design will have to be defined in the development, as well as the implementation of the information ingestion, automation, and maintenance processes, and the entire necessary analytics ecosystem around the data lake. The platform as a service should be considered as the main contribution of this work (Figure 2). The architecture proposed in Figure 3 offers two types of DSSs: the first allows for the visualization of data as a map or graph, and the second aims to anticipate what will happen (predictive). It is in this second phase that AI algorithms come into play, as they are able to learn from all of the historical information.

The sequence of DSS included the following basic steps (Figure 3):

- Import data raw from several sources;
- Data cleaning and standardization, and combination of data sources;
- Develop a predictive model based on the aggregated data, and by using DSS tools and ML techniques;
- Data exploitation through tables, graphs, alerts, or predictions.

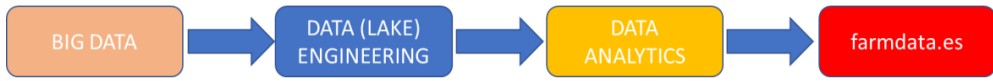

**Figure 2.** Technical digital platform system architecture.

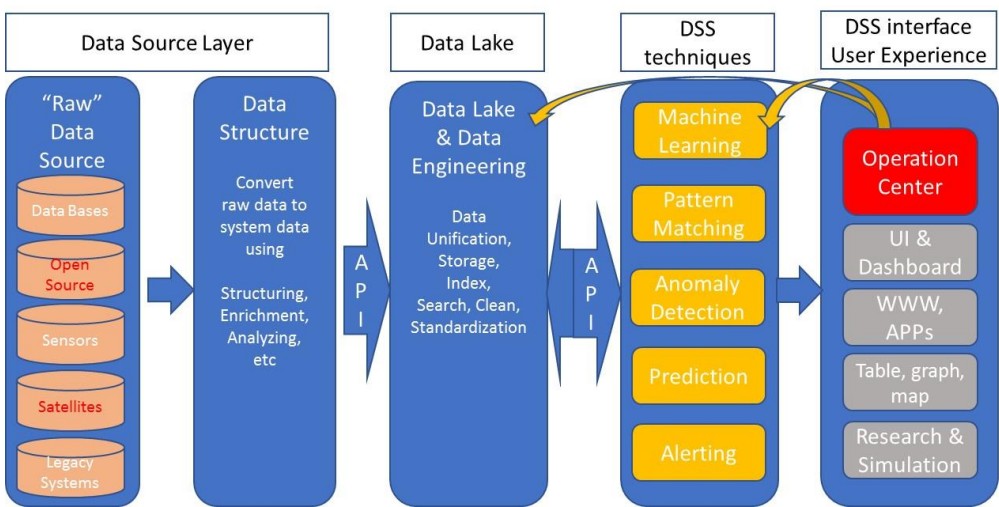

**Figure 3.** Socio-technical digital platform system architecture.

The modular design of the platform allows for additional programs and services to be added and interfaced with it. Private agents or commercial application developers can take advantage of the platform to develop new products or to offer complementary services to the farmer, on an agreement basis, allowing them to access market segments more quickly.

Finally, the frontend consists of a single-page application that has been developed within the framework of the AGROMIS I project (Figure 4).

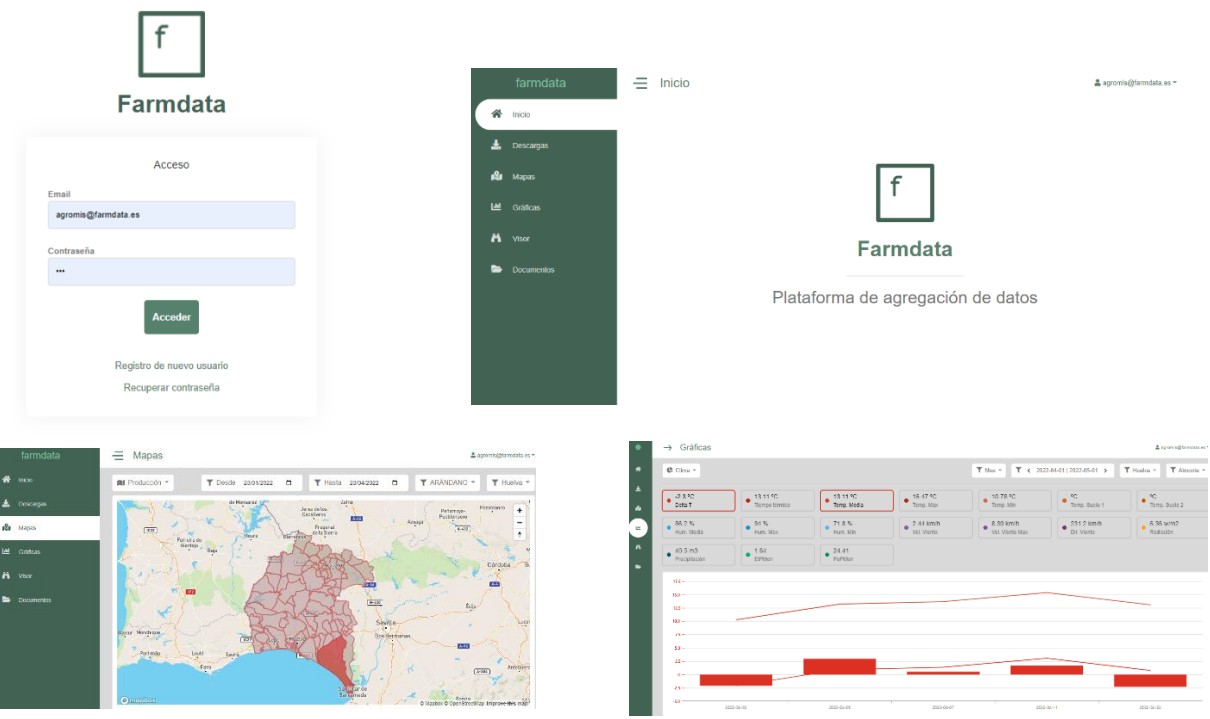

**Figure 4.** https://app.farmdata.es (accessed on 25 March 2022) digital platform frontend.

## 4.2. Data Platform Governance

Following the process of conducting the interviews and their subsequent analysis, some elements were identified as being essential for their incorporation into the design of the data platform. These included those that allow for the creation of an environment of trust and transparency, the legal terms and conditions of use, and the incentive system (Table 3).

**Table 3.** Questionnaire and responses *.

| Category and Statements | Mean | SD. |
|---|---|---|
| 1. Ownership, Access, Control | | |
| 1.1. I consider the data farm as belonging to the farmer | 4 | 1 |
| 1.2. The use of a platform service means that ownership is transferred to the service provider | 3.1 | 1.1 |
| 1.3. Data sharing with a government agency does not mean that it is public information | 3.6 | 0.99 |
| 1.4. The data have to include a description of who owns the data and the original source | 3.2 | 1.04 |
| 1.5. Privacy is important to me | 4.6 | 0.87 |
| 1.6. I do not feel in control of the data I share | 4 | 0.97 |
| 1.7. I do not feel comfortable sharing data from my farm | 4.2 | 0.85 |
| 2. Legal Considerations and Terms | | |
| 2.1. There should be a license on who can access and use the data | 4.1 | 1.05 |
| 2.2. There should be a license on the terms of use | 4.2 | 0.85 |
| 2.3. The terms of use have to be easy to identify, read, and understand | 3.7 | 0.99 |
| 2.4. I allow the commercial use of the data | 4 | 0.85 |
| 2.5. A person or organization can reuse the data to make money | 4 | 0.97 |
| 2.6. The data exchange model should be clearly explained to all actors involved | 3.5 | 0.9 |
| 2.7. The license agreement should be monitored or audited by a third party | 4.5 | 0.92 |
| 3. Perceived Benefits and Threats | | |
| 3.1. I consider there are more advantages than disadvantages of sharing data with the platform | 4.3 | 1.1 |
| 3.2. My perception of the benefits from using this platform is positive | 3.5 | 0.99 |
| 3.3. I like to get monetary benefits from the platform | 4.2 | 0.95 |
| 3.4. I value receiving any free features or services | 4 | 0.85 |
| 3.5. I feel like I belong to a specific network or community | 3.5 | 0.87 |
| 3.6. There are some factors that may limit my willingness to share data with this platform | 3.8 | 0.92 |
| 3.7. I am concerned about sharing data with this platform | 4.5 | 0.89 |

* 5—Point Likert scale, with 5 being strongly agree.

Thus, the following information was provided in the documentation section of the platform:

- Trust and transparency
    - Clear conditions, contractual or legal, regarding who has access to use the data;
    - Mechanisms that ensure secure data storage and management;
    - Information on how the data will be used and how the value generated will be distributed;
    - Guarantees that the service provider can be visibly identified, and its credibility verified.

- Terms and conditions of use
    - Customized and precise descriptions of ownership rights, conditions of access, and use covering raw data and derivative data;
    - Specific descriptions on who has access to what part of the information, and for what purpose;
    - Information on the data-sharing approach, from closed to open models.

- Incentives
    - Definition of the type of incentives to be offered for the provision of private data, in order to have a fair scenario for all interested parties, including commercial reuse.

In the same way, the design of our platform had to address the following concerns, as previous identified in the interviews:

- Data preservation: Some data may be retained for long-term use and analysis, requiring a highly centralized and reliable platform that is managed by specialized managers. It may become necessary to provide for a scalable design according to future requirements and needs for platform growth;

- Data integration: Even if private data is shared, it is not certain whether it is compatible with other private or public data sets. It was necessary to foresee bidirectional communication standards;
- Security: Security had to be addressed to eliminate farmers' concerns about sharing their data;
- Data standardization: Standardization mechanisms for metadata were contemplated to support Big Data analytical methods, such as data mining;
- Data scarcity: Crucial data remains scarce in many areas such as water use, carbon footprint, or household outcomes). The platform should consider using alternative data sources such as remote sensing imagery with UAVs and field sensors, or user behavior from social network sites to incorporate into its predictive themes;
- Data monetization: Farmers will pay for digital services if they receive valuable personalized advice related to their data exploitation. It is considered that some digital services can be provided free of charge, while others can be offered on a pay-per-use basis. In the latter case, the purchase of data will constitute a low cost for the platform.

It emerged from the participants' responses that the issue of controlling who has access to the data is seen as a right of the data producer. It was also interesting to note that the pilot participants thought that the owner's right of control should not be considered an extra payment feature, and that sharing private data with the public sector is not entirely convincing, due to potential audits of the information provided.

Additionally, the governance model for the digital platform had to address the following issues:

- Which model would generate trust and security for data sharing?
- Which model could more easily integrate a larger number of producers?
- Which model could provide innovative solutions on a more continuous and sustainable basis?
- Which model would best help or solve farmers' problems?

From the point of view of multiple stakeholders, combining data on a platform such as farmdata will require the willingness of farmers and management to share data, of researchers to build knowledge, of spin-offs to transfer new knowledge to the market, plus data management professionals to help overcome technical barriers [87]. In addition, governments should promote information exchange. Farmers can provide feedback for any improvements that additions to the platform may provide to increase performance [88]. Farmer organizations, such as cooperatives, can further promote the professionalization of their farmers by facilitating the acquisition of digital skills. Research institutions can develop key technologies, and commercial service providers can develop complementary services.

Although farmers contribute raw data, they have no right to collection and analysis [2], and no control over the distribution of the revenue generated from the data. We consider that the free apps provided by complementors could lead to farmers towards sharing data. This result is consistent with previous work, which argues that individuals who have stated strong privacy preferences are often willing to trade in their privacy for relatively small rewards [89].

The transparency of data use is unclear, and farmers are unwilling to share their data [12] unless they understand what is being done with it [16,39,40]. Given the intrinsic costs of collaboration and the many disincentives for farmers to treat agricultural data as a public good, public institutions have a key role to play in realizing this vision, by creating an environment that favors the sharing of data among the agricultural sector [1]. It is therefore essential that the European Commission develops its European data strategy [90] to ensure that data are stored securely, and that they are clearly and reliably exploited for the benefit of citizens, businesses, researchers, and public administrations. The General Data Protection Regulation [91] and the proposal on privacy and electronic communications, ePrivacy Regulation 2021 [92], will protect individuals within the EU from third-party intrusion into their private communications, unless they give their prior consent. As the agricultural sector is less data-aware compared to other sectors, there is a need to systematically initiate

a data-oriented dialogue with farmers. It should create pre-competitive data sharing spaces and infrastructure that is equally open to all stakeholders. Regulation can play an important role here, introducing contractual conditions that protect the interests of farmers, but that also give opportunities to companies and research organizations to innovate.

Finally, farmers are willing to share their data with a university-run platform, as it may be considered less risky, as universities do not have a commercial or regulatory relationship with farmers, in line with [24], where farmers would join a centralized Big Data platform, but only one that is managed by a university.

## 5. Conclusions

The result of the case study (farmdata) is a digital platform on a SQL database that is hosted on AWS, which uses cloud computing and storage services, interacts directly with Mapbox [93] for data visualization, and allows for retrieval through an easy-to-use control panel. Farmdata connects to several regional and national public and private databases. Both the public open data and private data from apps are aggregated in raw form into the farmdata data lake. University spin-offs and other approved providers can interact with the platform by contributing tools and by developing complementary services using data from the platform, thereby increasing its impact for farmers and other stakeholders [94].

In this document, we define AgDP, explain its operating principles and characteristics, conduct a case study of a digital platform for the Andalusian fruit and vegetable sector, and propose its governance model. We also analyze the technology that is required for the operation of the system, and describe the current application of these technologies. The case study research introduced and tested in the AGROMIS I project is a good starting point that can be adapted and replicated at regional and national levels.

Although we value its usefulness, we are aware of some problems. The creation of monopolies within these data ecosystems, and the excessive concentration of power by large companies is a potential threat that may be avoided by a university run-platform. The governance model that is proposed in this case study should combine centralized control, preferably management by universities rather than by governments, associations, or private companies, with a distributed organization model to ensure transparency. The potential size of this kind of platform will depend on farmer interests, which will be influenced by the benefits gained versus the cost of sharing private data. We believe that the solution to these problems of agricultural digital platforms requires the joint efforts of the government, the agricultural sector, ancillary companies, and research centers. The government should provide policy support and develop legislation for data security, agribusinesses should increase trust among themselves, and research institutes should provide innovations.

An additional challenge is the lack of digital skills among farmers. Farmers need basic skills that enable them to use digital tools, such as the applications and platforms that have been developed for farmers. To achieve this goal, it is necessary to integrate a government-subsidized digital agriculture program into university extension courses for both farmers and technicians, that are managed by cooperatives and agricultural organizations [95].

To address the challenges and opportunities described above, interdisciplinary collaboration between the research communities and the agricultural sector is also necessary. The development of IoT and AI hardware at universities will be another core component of this collaboration.

The platform designed and developed for the Andalusian fruit and vegetable sector is not a recipe that can be applied as such to all agricultural sectors. In the future, it is important to study how to solve these problems for an AgDP, which requires the joint efforts of the government, researchers, farmers, and companies in this direction.

As for future directions, we point to two: The first is with regard to platform sustainability. This raises the need for understanding and evaluating the role of platform governance and incentives [41,96], which requires further research on how the platform owner can influence value creation mechanisms in the digital platform ecosystem, how value is distributed between the platform owner and the complementors [97], and how

farmers sharing their data on the digital platform can influence the value creation mechanisms and grow the platform. A second area of future research is related to the achievement of a transparent and reliable ecosystem that can be solved with the application of the emerging blockchain phenomenon [98], which will then lead to the study of the fully distributed organization model.

**Author Contributions:** Conceptualization, J.D.B.; methodology, J.D.B.; software, J.M.; validation, J.D.B.; formal analysis, J.D.B.; investigation, J.D.B.; resources, J.D.B.; data curation, J.M.; writing, J.D.B.; visualization, J.D.B.; supervision, J.D.B.; funding acquisition, J.D.B. All authors have read and agreed to the published version of the manuscript.

**Funding:** This work has been supported by the AgroMIS Project, funding by the Regional Ministry of Economic Transformation, Industry, Knowledge and Universities, of the Government of Andalusia and European Regional Development Fund (ERDF). Code: A1122062E0_AgroMIS. University of Huelva and ceiA3.

**Institutional Review Board Statement:** Not applicable.

**Informed Consent Statement:** Not applicable.

**Data Availability Statement:** Not applicable.

**Acknowledgments:** The authors would like to thank Bo True Activities for providing their apps for their participation in the case study.

**Conflicts of Interest:** The authors declare no conflict of interest, and the funders had no role in the design of the study; in the collection, analyses, or interpretation of data; in the writing of the manuscript, or in the decision to publish the results.

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
