# Peer review of "A Case Study of a Digital Data Platform for the Agricultural Sector: A Valuable Decision Support System for Small Farmers"

_agriculture, doi:10.3390/agriculture12060767_

Round 1

Reviewer 1 Report

Dear Authors, 

The paper is well written. As far as its improvement in terms of appearance: 

  1. Try to use as uniformly as possible paragraph length. In some cases you have long paragraphs and in other very short ones.
  2. Please try to better highlight the contribution and novelty of this paper. In my opinion it is not clearly stated. A DSS with ML capabilities is very strong. However, similar systems are already there. 

Overall the paper is solid.

Author Response

Thank you for your dedication and advices.

  1. Try to use as uniformly as possible paragraph length. In some cases you have long paragraphs and in other very short ones.
    1. R: Thank you. We have revised the manuscript again and now the length of the paragraphs is similar.
  2. Please try to better highlight the contribution and novelty of this paper. In my opinion it is not clearly stated. A DSS with ML capabilities is very strong. However, similar systems are already there.
    1. R: Thanks for your suggestion. In reviewing the manuscript we have paid attention to the contribution and novelty. See introduction and specifically lines 44-45, 58-60, 81-84 and 90-98

Reviewer 2 Report

  1. In the title of the paper, it is very long; it needs to be shortened
  2. The title does not match the content of the paper. In the paper, the authors describe the case study while the title refers to the design and implementation of the study.
  3. Can you please modify the title according to the paper?
  4. ML is abbreviated in the abstract, but not defined. Please present the full name first and then use the abbreviation ML
  5. There in lines in the paper while the paper abstract concludes with the lines
  6. The keywords big-data is not used anywhere in the paper
  7. Please verify the keywords "decision support systems" as well
  8. The introduction describes the agricultural digital system, with references to the update, but there is no discussion of the implementation of the system.
  9. Perhaps it would be better to provide the table in a publication or statistical report. 
  10. I am not sure what the title of table 1 means.
  11. The section Method in that table does not match the sub-sections of section 3.
  12. It appears that no results were found in the results section. This paper needs to be improved and modified in accordance with the sections described.
  13. The paper does not have a research gap in it.
  14. There is no match between the conclusion of the abstract and the conclusion of the conclusion. The conclusion should be specific to the title and the abstract
  15. The references are good and are up-to-date
  16.  

Author Response

Thank you for your dedication and advices.

  1. In the title of the paper, it is very long; it needs to be shortened. The title does not match the content of the paper. In the paper, the authors describe the case study while the title refers to the design and implementation of the study. Can you please modify the title according to the paper?
    1. R: Thank you for your comment. We have proceeded to modify the title. Now is shorter and fits better with the content.
  1. ML is abbreviated in the abstract, but not defined. Please present the full name first and then use the abbreviation ML
    1. R: Thanks. When reviewing the abstract we have removed the word ML. However, in the text the first time an acronym is used it is defined the first time (see decision support systems (DSSs) L13 or machine learning (ML) in L39.
  2. There in lines in the paper while the paper abstract concludes with the lines
    1. R: We are sorry but we do not understand this comment. However, we have revised the paper and abstract and I hope that these changes will resolve the issue.
  3. The keywords big-data is not used anywhere in the paper. Please verify the keywords "decision support systems" as well
    1. R: The term big data appears in the text 22 times (though admittedly not in the abstract) and decision support system (DSS) thirteen times. If you have no objection, we have not removed them.
  4. The introduction describes the agricultural digital system, with references to the update, but there is no discussion of the implementation of the system. Perhaps it would be better to provide the table in a publication or statistical report
    1. R: Thank for your comment. In the introduction and literature we describe the digital technologies up to the digital platforms. We end with the identification of the main digital platforms for agriculture (Table 1), describing them as far as the object of the work is concerned (governance, ownership, ecosystem and services). Perhaps we have not seen it possible to provide the table as a statistical publication or report. We hope that the changes introduced in the introduction and literature sections of this paper will help to provide a more comprehensive picture of the digital platforms for agriculture.
  5. I am not sure what the title of table 1 means.
    1. R: Thanks. We have modified the title of the Table 1. We hope to make its meaning clearer.
  6. The section Method in that table does not match the sub-sections of section 3.
    1. R: Thank you. You are right. We have proceeded to align objectives (with research questions), methods and results. We have also reduced the Methods sections from three to two.
  7. It appears that no results were found in the results section. This paper needs to be improved and modified in accordance with the sections described.
    1. R: Thank for your comment. As you will see, the paper has been completely revised and improved. The results section now shows the results of the case study, with two subsections, each answering a research question.
  8. The paper does not have a research gap in it.
    1. R: Thank you. the previous version of the article did not correctly display unresolved issues. We have rewritten the introductory section identifying research gaps (in line 83 we refer to them) and formulated two more appropriate research questions.
  9. There is no match between the conclusion of the abstract and the conclusion of the conclusion. The conclusion should be specific to the title and the abstract
    1. R: After reviewing the title, abstract, introduction, methods, results and conclusions, we now think that all these sections are appropriately aligned.
  10. The references are good and are up-to-date
    1. R: Thank. We have include some new references.

Reviewer 3 Report

This is a good literature review. The authors refer an amount of literature and summarize the key requirements of decision support systems for small farms: smart irrigation decision, crop nutrient management and  feeding decision making. They propose that the digital agriculture platform has three layers: data, infrastructure and network, and relies on three key technologies: big data, machine learning and cloud computing. The paper lists several typical digital farm management platforms and briefly analyzes their characteristics. The aim and design of the research, the design and management of the platform, and in particular the user's subjective views on data ownership, legitimacy, risk and benefit are discussed, which is instructive for other studies. But the article does not delve into the features and requirements of some specific technologies, such as machine learning, smart decision and NoSQL. In addition, it is recommended to reorganize the title of the body section, the content of the Method, Result and the title do not match, you can consider grouped by theme.

Author Response

Thank you for your dedication and advices.

  1. This is a good literature review.
    1. R: Thank you.
  2. The authors refer an amount of literature and summarize the key requirements of decision support systems for small farms: smart irrigation decision, crop nutrient management and feeding decision making. They propose that the digital agriculture platform has three layers: data, infrastructure and network, and relies on three key technologies: big data, machine learning and cloud computing. The paper lists several typical digital farm management platforms and briefly analyzes their characteristics. The aim and design of the research, the design and management of the platform, and in particular the user's subjective views on data ownership, legitimacy, risk and benefit are discussed, which is instructive for other studies. But the article does not delve into the features and requirements of some specific technologies, such as machine learning, smart decision and NoSQL.
    1. R: Thank you for your comment. You are right that the article does not delve (only identifies or describes) into the features and requirements of some specific technologies, such as machine learning, intelligent decision, and NoSQL. Although the article focuses more on design aspects, such as what elements the platform should have (RQ1) and qualitative ones, such as the type of governance (RQ2) than strictly technical or analysis, we have added new references in the sense of your suggestion. For ML see L70, L 73, L 201. See lines 71 for Smart decision (ref 29-31) or L276 for NoSQL among others.
  3. In addition, it is recommended to reorganize the title of the body section, the content of the Method, Result and the title do not match, you can consider grouped by theme.
    1. R: Thank you. We have proceeded to improve the paper and after reviewing the title, abstract, introduction, methods, results and conclusions, we now think that all these sections are appropriately aligned

Round 2

Reviewer 2 Report

The authors answered all my previous comments